# TOWARDS EXPRESSIVE GRAPH REPRESENTATIONS FOR GRAPH NEURAL NETWORKS

## ABSTRACT

Graph Neural Network (GNN) aggregates the neighborhood information into the node embedding and shows its powerful capability for graph representation learning in various application areas. However, most existing GNN variants aggregate the neighborhood information in a fixed non-injective fashion, which may map different graphs or nodes to the same embedding, detrimental to the model expressiveness. In this paper, we present a theoretical framework to improve the expressive power of GNN by taking both injectivity and continuity into account. Based on the framework, we develop *injective and continuous expressive Graph Neural Network* (iceGNN) that learns the graph and node representations in an injective and continuous fashion, so that it can map similar nodes or graphs to similar embeddings, and non-equivalent nodes or non-isomorphic graphs to different embeddings. We validate the proposed iceGNN model for graph classification and node classification on multiple benchmark datasets. The experimental results demonstrate that our model achieves state-of-the-art performances on most of the benchmarks.

## 1 INTRODUCTION

Graph representation learning that maps graphs or their components to vector representations has attracted growing attentions for graph analysis. Recently, graph neural networks (GNN) that can learn a distributed representation for a graph or a node in a graph are widely applied to a variety of areas, such as social network analysis (Hamilton et al., 2017; Ying et al., 2018a), molecular structure inference (Duvenaud et al., 2015; Gilmer et al., 2017), text mining (Yao et al., 2019; Peng et al., 2018), clinical decision-making (Mao et al., 2022b; Li et al., 2018) and image processing (Mao et al., 2022a; Garcia & Bruna, 2018). GNN recursively updates the representation of a node in a graph by aggregating the feature vectors of its neighbors and itself (Hamilton et al., 2017; Morris et al., 2019; Xu et al., 2019). The graph-level representation can then be obtained through aggregating the final representations of all the nodes in the graph. The learned representations can be fed into a prediction model for different learning tasks, such as node classification and graph classification.

In GNN, the aggregation rule plays a vital role in learning expressive representations for the nodes and the entire graph. There are many GNN variants with different aggregation rules proposed to achieve good performances for different tasks and different problems (Kipf & Welling, 2017; Hamilton et al., 2017; Zhang et al., 2018; Xinyi & Chen, 2019; Wang et al., 2020). However, most of the existing GNN aggregation rules are designed based on a fixed non-injective pooling function, (e.g., max pooling and mean pooling) or on non-continuous node types (e.g., graph isomorphism test). The non-injective aggregation may map different (non-isomorphic) graphs or (non-equivalent) nodes to the same embedding; and the non-continuous aggregation may map similar graphs or nodes to quite different embeddings, both detrimental to the expressive power of GNN. For example, for the graph with attributed nodes in Figure 1(a), mean pooling or sum aggregation on the neighborhoods generates the same neighborhood representation for all the nodes (Figure 1(d)), thus cannot capture any meaningful structure information. Xu et al. (2019) showed that a powerful GNN can at most achieve the discriminative power of Weisfeiler-Lehman graph isomorphism test (WL test) which can discriminate a broad class of graphs (Weisfeiler & Lehman, 1968), and proposed the powerful graph isomorphism network (GIN). However, the theoretical framework of GIN is under the assumption that the input feature space is countable, which makes GIN less expressive when applied to graphs with continuous attributes.

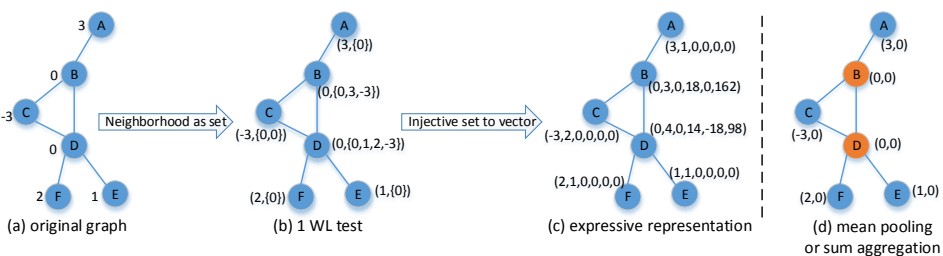

Figure 1: An overview of our framework on an exemplar attributed graph in one iteration. (a) Original graph with attributed nodes; (b) Graph nodes are represented by the corresponding attribute and neighborhood set through WL test; (c) The node vector representations after an injective set function on neighborhood sets, here the set function is $f(X) = \sum_{x \in X}(1, x, x^2, x^3, x^4)$; (d) A non-injective alternative set function in GNNs, after aggregation, the node information remain unchanged, node B and D still have the same representation despite their different neighborhoods.

We argue that the expressive power of a graph mapping should imply two aspects, injectivity and continuity: the injectivity ensures different graphs are mapped to different representations and the continuity ensures that similar graphs are mapped to similar representations. Most previous works only took either one into account for GNN design; few considered both injectivity and continuity. Here, we present a theoretical framework that can guide us to design highly expressive GNNs with both injectivity and continuity for general graphs with continuous attributes. We also present a necessary condition related to the representation dimension for a fully injective and continuous graph mapping. The general idea of our framework is illustrated in Figure 1.

Our main contributions are summarized as follows. (1) We present a theoretical framework to guide the design of expressive GNNs by ensuring the injectivity and continuity in the neighborhood aggregation process. (2) We present a limitation about the representation dimension for a fully injective and continuous graph mapping. (3) Based on the framework, we implement two injective and continuous expressive GNN (iceGNN) models with a fixed and learnable aggregation function, respectively. (4) We validate our models on multiple benchmark datasets including simple graphs and attributed graphs for graph classification and node classification, the experimental results demonstrate that our models can achieve state-of-the-art performances on most of the benchmarks. Our code is available in the Supplementary Material. Common notations used throughout the paper are found in Appendix A.1 Table 5.

## 2 RELATED WORK

Many GNN variants with different aggregation rules are proposed in the literature to achieve good performances in different tasks. GIN proposed by Xu et al. (2019) is expected to be highly expressive for simple graphs where node attributes are one-hot encoders on which sum aggregation is injective. However, GIN cannot be directly extended to attributed graphs with the same expressive power, because the sum aggregation is no longer injective in uncountable cases. GCN is another GNN variant with simple element-wise mean pooling in a node's neighborhood, including the node itself (Kipf & Welling, 2017). Hamilton et al. (2017) tested 3 aggregators in GraphSAGE, including mean aggregator, LSTM aggregator and max pooling aggregator, they found no significant performance difference exists between the LSTM aggregators and pool aggregators, but GraphSAGE-LSTM is significantly slower than GraphSAGE-pool. Mean aggregation and max pooling are permutation invariant on sets, but the operation is not injective, which may result in the same embedding for different inputs. LSTM aggregation could have large expressive capacity, but it is not permutation invariant, this may cause equivalent nodes or isomorphic graphs to have different embeddings. Corso et al. (2020) combined multiple aggregators with degree-scalers and proposed PNA to improve the expressive power of GNN, but they did not provide a theoretical guidance on how to improve the expressive power of GNN. In this article, we present a theoretical framework to guide the design of expressive GNNs by ensuring the injectivity and continuity in the neighborhood aggregation process. PNA can exactly fall into our framework by a simple comparison analysis.

Theoretical studies on GNNs showed the expressive power of GNNs has been linked to the WL test (Morris et al., 2019; Xu et al., 2019). Xu et al. (2019) showed that GNNs with 1-hop neighborhood aggregation can at most achieve the expressive power of the 1-WL test, and developed GIN that can achieve this expressive power in countable space. Based on k-WL tests, Morris et al. (2019) proposed k-GNN, which can take higher-order interactions among nodes into account. Maron et al. (2019a) proved that order-k invariant graph networks are at least as powerful as the k-WL tests and developed a GNN model that is more powerful than message passing GNNs, possessing the expressiveness of 3-WL, but the higher expressive power comes with a computational cost of processing high order tensors. A survey on the expressive power of GNN can be found in Sato (2020).

## 3 PRELIMINARIES

### 3.1 GRAPH NEURAL NETWORKS

Most modern GNNs fall into the category of message passing neural networks (Gilmer et al., 2017) that follow a neighborhood aggregation strategy that recursively updates a node representation by aggregating representations of its neighbors and the node itself. The graph-level representation is obtained through aggregating the final representations of all the nodes in the graph. Formally, the propagation rule of a GNN layer can be represented as

$$H_{\mathcal{N}(v)}^{(k)} = f_A^{(k)} \left( \left\{ H^{(k)}(w) | w \in \mathcal{N}(v) \right\} \right) ; \tag{1}$$

$$H^{(k+1)}(v) = f_C^{(k)} \left( H^{(k)}(v), H_{\mathcal{N}(v)}^{(k)} \right) \tag{2}$$

where $H^{(k)}(v)$ is the representation vector of node $v$ in the $k$th layer, and $H^{(0)}(v)$ is initialized with $X(v)$, the original attributes of node $v$. $\mathcal{N}(v)$ is the neighborhood of $v$. $f_A^{(k)}(\cdot)$ aggregates over the neighborhood $\mathcal{N}(v)$ to generate a neighborhood representation $H_{\mathcal{N}(v)}^{(k)}$, and $f_C^{(k)}(\cdot)$ combines the node's current representation $H^{(k)}(v)$ and its neighborhood's representation $H_{\mathcal{N}(v)}^{(k)}$ in the $k$th layer.

For node embedding, the node representation in the final layer $H^{(K)}(v)$ (suppose a total of $K$ layers) is considered as an informative representation that could be used for downstream tasks, e.g., node classification. For graph or subgraph embedding, another aggregation function $f_R(\cdot)$ is employed to obtain the graph-level representation $h_G$ by aggregating the final representations of all nodes in the graph or subgraph $G$, i.e.,

$$H_G = f_R \left( \left\{ H^{(K)}(v) | v \in G \right\} \right) \tag{3}$$

$f_A(\cdot)$, $f_C(\cdot)$ and $f_R(\cdot)$ are all crucial for the expressive power of a GNN. $f_A(\cdot)$ and $f_R(\cdot)$ are set functions that map a set to a vector, they can be simple summations or sophisticated graph-level pooling functions (Ying et al., 2018b; Zhang et al., 2018; Wang et al., 2020). $f_C(\cdot)$ operates on two vectors, it can be usually modeled by a multi-layer perceptron (MLP) or linear function on the concatenated vector.

### 3.2 THE EXPRESSIVE POWER OF GNN

Recently, theoretical analysis showed that the expressive power of a GNN is associated with the WL test (Morris et al., 2019; Xu et al., 2019). Xu et al. (2019) proved that the expressive power of GNN is bounded by the one-dimensional Weisfeiler-Lehman test. The following Lemma and Theorem from Xu et al. (2019) describe the relation between GNNs and WL test in expressive power on discriminating graphs, refer to Xu et al. (2019) for the proofs.

**Lemma 1.** *If the WL test decides two graphs $G_1$ and $G_2$ are isomorphic, any GNNs defined by Eq. 2 and 3 will map $G_1$ and $G_2$ to the same embedding.*

**Theorem 1.** *If WL test decides two graphs $G_1$ and $G_2$ are not isomorphic, a GNN with sufficiently many GNN layers defined by Eq. 2 and 3 can also map $G_1$ and $G_2$ to different embeddings if the functions $f_A(\cdot)$, $f_C(\cdot)$ and $f_R(\cdot)$ are all injective.*

The above Lemma and Theorem can guide us to design a GNN that has the discriminative power equal to WL test. The key is to design injective functions for $f_A(\cdot)$, $f_C(\cdot)$ and $f_R(\cdot)$. An injective function for $f_C(\cdot)$ that operates on two vectors can be easily obtained by concatenating the two vectors. But designing an injective function for $f_A(\cdot)$ or $f_R(\cdot)$ that operates on a set is not trivial, because sets can have different number of elements, and the operation on the set elements must be permutation-invariant.

While Xu et al. (2019) considered the injectivity in their framework, the continuity of these functions are also crucial to the model expressive capability, which was not considered in Xu et al. (2019). Some popular aggregations by pooling, e.g., mean pooling and max pooling, are continuous but not injective. Few works took both injectivity and continuity into account. In the following, we will present how to design the continuous injective aggregation function on a set and show a necessary condition related to the dimension for injective and continuous aggregation.

## 4 METHODS

### 4.1 SET REPRESENTATION

A set function is a function defined in a domain that is a collection of sets (specifically multisets where the same object can repeat multiple times, a set means a finite multiset all through the paper). In a finite graph, the neighborhood of each node is considered as a finite set. Thus, in this paper, we only consider set functions of finite sets. A continuous set function is of real importance in practice Wagstaff et al. (2019). The continuity of a function ensures that the change in output is very slight if the input is altered slightly by any reason, such as truncating to machine precision. In this paper, we consider the ordinary continuity, where the continuity of function $f(\mathbf{x})$ at point $\mathbf{c}$ is defined by the limit as $\lim_{\mathbf{x}\to\mathbf{c}} f(\mathbf{x}) = f(\mathbf{c})$.

For $M \in \mathbb{N}$, a set function $f(X)$ defined in domain $\mathcal{X}_M = \{X | X \subset \mathbb{R}^d, |X| \leq M\}$ can be represented as a sequence of permutation-invariant functions $f_i$ for different set sizes, i.e.,

$$f(X) = f_i(x_1, \cdots, x_i) \quad if \quad |X| = i \leq M, \tag{4}$$

where $x_1, \cdots, x_i \in X$.

**Definition 1** (Continuous set function). *For $M \in \mathbb{N}$ and a set function $f(X)$ defined in domain $\mathcal{X}_M = \{X \mid X \subset \mathbb{R}^d, |X| \leq M\}$, $f(X)$ is represented as Eq. 4, if $f_i(x_1, \cdots, x_i)$ is continuous in the Euclidean space for every $i \leq M$, we call $f(X)$ a continuous set function.*

Obviously, a continuous set function can also have the property that sufficiently small changes in the input (a sufficiently small change will not change the set size) result in arbitrarily small changes in the output. Thus, by a continuous set function, graphs with very similar structures and attributes could be mapped to similar embeddings. An injective set function can map distinct sets to distinct values. Thus, an injective and continuous set representation function can encode a set to a representation such that different sets have different representations and similar sets have similar representations.

The following theorem provides a way to construct continuous injective set functions in uncountable space by sum aggregation after a certain transformation.

**Theorem 2.** *Assume $\mathcal{X}_M$ is a set of finite subsets of $\mathbb{R}^d$ with size less than or equal to $M$, i.e., for $M \in \mathbb{N}$ and $\mathcal{X}_M = \{X | X \subset \mathbb{R}^d, |X| \leq M\}$, there exists an infinite number of continuous functions $\Phi : \mathbb{R}^d \to \mathbb{R}^D$ such that the set function $f : \mathcal{X}_M \to \mathbb{R}^D$, $f(X) = \sum_{\mathbf{x}\in X} \Phi(\mathbf{x})$ is continuous and injective.*

Theorem 2 generalizes Lemma 6 in Zaheer et al. (2017) to multidimensional cases. We prove Theorem 2 in Appendix A.2. The proof contains three steps: 1. Constructing a satisfying function in one dimensional cases ($d = 1$); 2. Constructing a satisfying function in multidimensional cases($d > 1$) based on the results in step 1; 3. The satisfying function can be used to generate infinitely many other satisfying functions.

In our proof, we find a $\Phi_M(\mathbf{x})$ defined in Eq. 5 (where $i, j = 1, \cdots, d$) that can make $f(X) = \sum_{\mathbf{x}\in X} \Phi(\mathbf{x})$ continuous and injective if the first entries of all vectors in $X$ are distinct. If the first entries of all vectors in $X$ are not distinct, we could add $P_i$ to $\Phi_M(\mathbf{x})$ (i.e., $\Phi'_M(\mathbf{x})$) to ensure fully injectivity and continuity.

$$
\begin{aligned}
P_i \quad &= [1, \mathbf{x}[i], \mathbf{x}[i]^2, \cdots, \mathbf{x}[i]^M] \\
P_{i,j} \quad &= [\mathbf{x}[j], \mathbf{x}[i]\mathbf{x}[j], \mathbf{x}[i]^2\mathbf{x}[j], \cdots \mathbf{x}[i]^{M-1}\mathbf{x}[j]] \\
\Phi_M(\mathbf{x}) \quad &= [P_1, P_{1,2}, \cdots, P_{1,d}] \\
\Phi'_M(\mathbf{x}) \quad &= [\Phi_M(\mathbf{x}), P_2, \cdots, P_d]
\end{aligned}
\tag{5}
$$

In the proof, we also provide a way to construct such a function $\Phi(\mathbf{x})$ by defining a continuous injective function $g : \mathbb{R}^d \rightarrow \mathbb{R}^d$, then $\Phi(g(\mathbf{x}))$ can also satisfy the condition if we have a function $\Phi(\mathbf{x})$ satisfying the condition. We call $\Phi(\mathbf{x})$ the **transformation function**.

The following theorem tells a necessary condition of constructing an injective and continuous set function by the sum aggregation.

**Theorem 3.** *Let $M \in \mathbb{N}$ and $\mathcal{X}_M = \{X | X \subset \mathbb{R}^d, |X| = M\}$, then for any continuous function $\Phi : \mathbb{R}^d \rightarrow \mathbb{R}^N$, if $N < dM$, the set function $f : \mathcal{X} \rightarrow \mathbb{R}^N$, $f(X) = \sum_{\mathbf{x} \in X} \Phi(\mathbf{x})$ is not injective.*

We prove Theorem 3 in Appendix A.3. Theorem 3 tells that, to construct a continuous injective set function for sets with $M$ $d$-dimensional vectors by sum aggregation with continuous transformation $\Phi(\mathbf{x})$, $\Phi(\mathbf{x})$ must have at least $dM$ dimensions. We are restricting $\Phi$ as a continuous function so that it can be modeled by a neural network, because a neural network can approximate any continuous function rather than any function by the universal approximation theorem (Cybenko, 1989).

## 4.2 INJECTIVE AND CONTINUOUS EXPRESSIVE GRAPH NEURAL NETWORKS

Since Theorem 2 tells that a set can be uniquely represented by a sum aggregation of its elements through a continuous transformation function, we can use the unique set representation to model the neighborhood of each node in a graph, and thus improve the expressiveness of graph representation. From Theorem 1, to design an expressive GNN, we need to design injective and continuous functions for $f_A(\cdot)$, $f_C(\cdot)$ and $f_R(\cdot)$. Since $f_C(\cdot)$ is easy to get continuous and injective, and $f_A(\cdot)$ and $f_R(\cdot)$ both operate on a set of vectors in $\mathbb{R}^d$, thus, we need Theorem 2 to guide us to construct a continuous and injective set function for $f_A(\cdot)$ and $f_R(\cdot)$ by sum aggregation after a certain continuous transformation.

**COMBINE function.** According to Theorem 3, for a set of $M$ $d$-dimensional embeddings, the transformation function must be at least $dM$-dimensional to construct a continuous injective set function with sum aggregation, thus, without dimension reduction in $f_C(\cdot)$, we get a $(dM + d)$-dimensional embedding after one layer ($dM$ for the set of neighbors and $d$ for a node's current dimension). After $k$ layers, the output embeddings have $d(M + 1)^k$ dimensions, which makes it impractical to implement a fully injective and continuous GNN for large graphs. Nevertheless, in a specific learning task, not all dimensions are related to the learning task, we could design learnable neural networks (e.g., MLP) to adaptively reduce the output dimension in each layer as Eq. 6.

$$
f_C^{(k)}(\mathbf{x_1}, \mathbf{x_2}) = MLP^{(k)}([\mathbf{x_1}, \mathbf{x_2}])
\tag{6}
$$

Note that an MLP mapping high-dimensional vectors to low-dimensional vectors cannot be continuous and injective. Here, MLP is used for task-driven feature reduction.

**AGGREGATE function.** $f_A(\cdot)$ operates on a set of node embeddings in the neighborhood of a node. We have two choices of the transformation function of $f_A(\cdot)$, i.e., *fixed transformation* and *learnable transformation*.

*Fixed transformation.* In the proof of Theorem 2, we find the function $\Phi'_M(\mathbf{x})$ defined in Eq. 5 can be used as a continuous transformation function to make the sum aggregation continuous and injective in most cases. Let $M_n$ be the max neighborhood size for all nodes in all the graphs. Usually, if $M_n$ is not very large, we can set the transformation function as $\Phi'_{M_n}(\mathbf{x})$ for each layer $k$ to maintain the expressive power. Then $f_A(\cdot)$ for layer $k$ can be represented as

$$
f_A^{(k)}\left(\left\{H^{(k)}(w) | w \in N(v)\right\}\right) = \sum_{w \in \mathcal{N}(v)} \Phi'_{M_n}\left(H^{(k)}(w)\right)
\tag{7}
$$

Combining Eqs. 1, 2, 6 and 7, we get the propagation rule,

$$H^{(k+1)}(v) = MLP^{(k)}([H^{(k)}(v), \sum_{w \in \mathcal{N}(v)} \Phi'_{M_n}(H^{(k)}(w))]) \tag{8}$$

Though the function $\Phi'_M(\mathbf{x})$ defined in Eq. 5 can make the sum aggregation continuous and injective, it may result in numerical stability since the item $\mathbf{x}[i]^M$ will make the number become very large or very close to 0 if $M$ is very large. To address this issue, we use a continuous and injective function $g(\mathbf{x})$ to normalize the power, since in the proof of Theorem 2 we know $\Phi_M(g(\mathbf{x}))$ is also a qualified transformation function to make the sum aggregation continuous and injective, if $g(\mathbf{x})$ is continuous and injective. In this paper, we set

$$g(\mathbf{x})[i] = \begin{cases} \mathbf{x}[i]^{1/M}, & \mathbf{x}[i] \geq 0 \\ -(-\mathbf{x}[i])^{1/M}, & \mathbf{x}[i] < 0 \end{cases} \tag{9}$$

*Learnable transformation.* Due to the continuity of the transformation function, we can also set a learnable MLP to approach the transformation function for $f_A(\cdot)$ by the universal approximation theorem (Cybenko, 1989), then we get the propagation rule as

$$H^{(k+1)}(v) = MLP_c^{(k)}([H^{(k)}(v), \sum_{w \in \mathcal{N}(v)} MLP_t^{(k)}(H^{(k)}(w))]) \tag{10}$$

where $MLP_t^{(k)}$ and $MLP_c^{(k)}$ serve as the transformation function and the combine function for the $k$th layer, respectively. By this way, we get all the node embeddings for all graphs. For graph or subgraph embedding, we need another aggregation function $f_R(\cdot)$ to aggregate all the node embeddings in a graph.

**READOUT function.** $f_R(\cdot)$ operates on a set of all node embeddings in a graph. For large graphs with many nodes, a fully injective and continuous set function will generate a high-dimensional embeddings. We also use a learnable MLP as the transformation function to reduce the output dimension.

$$H_G = f_R\left(\left\{H^{(K)}(v)|v \in G\right\}\right) = \sum_{v \in G} MLP_G\left(H^{(K)}(v)\right) \tag{11}$$

Note that the final node embeddings are output from an $MLP_c^{(K)}$ and directly input to $MLP_G$, we merge the two MLPs as one in the implementation. For graph classification, the output graph-level embedding are input to an MLP classifier with $n_C$ outputs corresponding to the probabilities of the $n_C$ classes. We only use the final GNN layer outputs for classification rather than concatenating all layers' outputs to construct a longer vector representation for classification as GIN did, because we think the final layer outputs contain all information from middle layers and are expressive enough for graph classification. In addition, this can reduce the input dimension of the final classifier, resulting in a simpler classifier than GIN, especially in case of many layers.

**Remark.** By the propagation rule in Eqs. 8 and 10, we implement two variants of iceGNNs, namely iceGNN-fixed and iceGNN-MLP, respectively. In practice, for a specific learning task, e.g., graph classification that maps a graph to a single label, the whole process cannot be injective, and a certain dimension reduction process must be applied. The key is to ensure the dimension reduction is guided by the target task. For example, the sum aggregation is not injective, and moreover, the process is fixed and cannot be adjusted by the loss function; thus, the aggregation could lose important information that is related to the target task. In our framework, according to our theoretical result in Theorem 2, we could achieve an injective and continuous aggregation by employing a transformation function $\Phi(\mathbf{x})$. For iceGNN-fixed, a fixed transformation function is applied to make the aggregation injective and continuous, and then a learnable MLP is applied to reduce the dimension. And for iceGNN-MLP, we combine the aggregation and dimension reduction (linear function $L$) in one learnable MLP. Thus, the learned low-dim features are related to the task.

## 5 EXPERIMENTS

### 5.1 GRAPH CLASSIFICATION

**Datasets.** We use 8 simple graph benchmarks and 5 attributed graph benchmarks for graph classification, the 8 simple graph datasets contain 4 bioinformatics datasets (MUTAG, PTC, NCI1, PROTEINS)

Table 1: Accuracy for simple graph classification in test set (%). Top 3 performances on each dataset are bolded. The best performances are underlined. The first two rows are our results, the middle part corresponds the deep learning methods, the bottom part corresponds to the graph kernel methods.

| | MUTAG | PTC | NCI1 | PROTEINS | COLLAB | IMDB-B | IMDB-M | RDT-B |
|---|---|---|---|---|---|---|---|---|
| iceGNN-fixed | **91.1 ± 6.7** | **67.9 ± 7.3** | 82.9 ± 1.6 | **77.5 ± 6.2** | – | – | – | – |
| iceGNN-MLP | **90.6 ± 7.9** | **68.8 ± 7.2** | 83.6 ± 1.9 | 76.5 ± 5.5 | 78.7 ± 1.5 | 72.8 ± 3.9 | 50.3 ± 2.3 | **92.0 ± 3.5** |
| GIN-final | 90.0 ± 5.4 | 65.9 ± 6.1 | 81.4 ± 1.6 | 76.2 ± 4.9 | 75.2 ± 2.0 | 72.5 ± 3.9 | 48.9 ± 2.7 | **90.1 ± 5.3** |
| GIN (Xu et al., 2019) | 89.4 ± 5.6 | 64.6 ± 7.0 | 82.7 ± 1.6 | 76.2 ± 2.8 | **80.2 ± 1.9** | **75.1 ± 5.1** | **52.3 ± 2.8** | **92.4 ± 2.5** |
| GCN (Kipf & Welling, 2017) | 87.8 ± 6.0 | 62.7 ± 8.0 | 73.5 ± 1.4 | 71.0 ± 5.0 | 67.0 ± 3.6 | 71.3 ± 4.2 | 42.6 ± 5.2 | 65.1 ± 14.0 |
| GraphSAGE (Hamilton et al., 2017) | 85.1 ± 7.6 | 63.9 ± 7.7 | 77.7 ± 1.5 | 75.9 ± 3.2 | – | 72.3 ± 5.3 | **50.9 ± 2.2** | – |
| PSCN (Niepert et al., 2016) | **92.6 ± 4.2** | 60.0 ± 4.8 | 78.6 ± 1.9 | 75.9 ± 2.8 | 72.6 ± 2.2 | 71.0 ± 2.2 | 45.2 ± 2.8 | 86.3 ± 1.6 |
| CapsGNN (Xinyi & Chen, 2019) | 86.7 ± 6.9 | - | 78.4 ± 1.6 | 76.3 ± 3.6 | 79.6 ± 0.9 | 73.1 ± 4.8 | 50.3 ± 2.7 | – |
| GCAPS-CNN (Verma & Zhang, 2018) | – | 66.0 ± 5.9 | 82.7 ± 2.4 | 76.4 ± 4.2 | 77.7 ± 2.5 | 71.7 ± 3.4 | 48.5 ± 4.1 | 87.6 ± 2.5 |
| IEGN (Maron et al., 2019b) | 84.6 ± 10 | 59.5 ± 7.3 | 73.7 ± 2.6 | 75.2 ± 4.3 | 77.9 ± 1.7 | 71.3 ± 4.5 | 48.6 ± 3.9 | – |
| 1-2-3 GNN (Morris et al., 2019) | 86.1 | 60.9 | 76.2 | 75.9 | – | **74.2** | 49.5 | – |
| HaarPool (Wang et al., 2020) | 90.0 ± 3.6 | – | 78.6 ± 0.5 | **80.4 ± 0.8** | – | – | – | – |
| 3WLGNN (Maron et al., 2019a) | 90.5 ± 8.7 | 66.2 ± 6.5 | 83.2 ± 1.1 | **77.2 ± 4.7** | **81.4 ± 1.4** | 73.0 ± 5.8 | 50.5 ± 3.6 | – |
| GHC (Nguyen & Maehara, 2020) | 89.3 ± 8.3 | – | – | – | – | 72.1 ± 2.6 | 48.6 ± 4.4 | – |
| Ring-GNN (Chen et al., 2019) | 78.1 ± 5.6 | – | – | – | – | 73.0 ± 5.4 | 48.2 ± 2.7 | – |
| InfoGraph (Sun et al., 2019) | 89.0 ± 1.1 | 61.6 ± 1.4 | – | – | – | 73.0 ± 0.9 | 49.7 ± 0.5 | 82.5 ± 1.4 |
| CMV-GR (Hassani & Khasahmadi, 2020) | 89.7 ± 1.1 | 62.5 ± 1.7 | – | – | – | **74.2 ± 0.7** | **51.2 ± 0.5** | 84.5 ± 0.6 |
| WL subtree (Shervashidze et al., 2011) | 90.4 ± 5.7 | 59.9 ± 4.3 | **86.0 ± 1.8** | 75.0 ± 3.1 | 78.9 ± 1.9 | 73.8 ± 3.9 | **50.9 ± 3.8** | 81.0 ± 3.1 |
| GK (Shervashidze et al., 2009) | 81.6 ± 2.1 | 57.3 ± 1.4 | 62.5 ± 0.3 | 71.7 ± 0.6 | 72.8 ± 0.3 | 65.9 ± 1.0 | 43.9 ± 0.4 | 77.3 ± 0.2 |
| DGK (Yanardag & Vishwanathan, 2015) | 87.4 ± 2.7 | 60.1 ± 2.6 | 80.3 ± 0.5 | 75.7 ± 0.5 | 73.1 ± 0.3 | 67.0 ± 0.6 | 44.6 ± 0.5 | 78.0 ± 0.4 |
| WL-OA (Kriege et al., 2016) | 84.5 ± 1.7 | 63.6 ± 1.5 | **86.1 ± 0.2** | 76.4 ± 0.4 | **80.7 ± 0.1** | – | – | 89.3 ± 0.3 |
| WWL (Togninalli et al., 2019) | 87.3 ± 1.5 | **66.3 ± 1.2** | **85.7 ± 0.2** | 74.2 ± 0.5 | – | – | – | – |
| MLG (Kondor & Pan, 2016) | 87.9 ± 1.6 | 63.3 ± 1.5 | 81.8 ± 0.2 | 76.3 ± 0.7 | – | 66.6 ± 0.3 | 41.2 ± 0.0 | – |

Table 2: Accuracy for attributed graph classification in test set (%). Top 3 performances on each dataset are bolded. The best performances are underlined.

| | ENZYMES | FRANKENSTEIN | PROTEINSatt | SYNTHETICnew | Synthie |
|---|---|---|---|---|---|
| iceGNN-fixed | 67.00 ± 6.40 | **74.59 ± 2.12** | **77.99 ± 1.77** | **97.67 ± 3.00** | **95.75 ± 2.25** |
| iceGNN-MLP | **71.50 ± 7.01** | 73.12 ± 2.59 | **77.27 ± 3.67** | **99.00 ± 1.53** | **99.25 ± 1.15** |
| GIN-final | 68.50 ± 4.86 | 68.66 ± 2.47 | 76.64 ± 2.94 | 83.33 ± 6.67 | 89.50 ± 2.18 |
| GCN (Kipf & Welling, 2017) | 44.17 ± 4.90 | 63.27 ± 1.15 | 68.83 ± 4.39 | 69.00 ± 7.00 | 49.25 ± 7.59 |
| HGK-SP (Morris et al., 2016) | 71.30 ± 0.86 | 70.06 ± 0.32 | **77.47 ± 0.43** | 96.46 ± 0.61 | 94.34 ±0.54 |
| HGK-WL (Morris et al., 2016) | 67.63 ± 0.95 | **73.62 ± 0.38** | 76.70 ± 0.41 | **98.84 ± 0.29** | **96.75 ± 0.51** |
| GHK (Feragen et al., 2013) | 68.80 ± 0.96 | 68.48 ± 0.26 | 72.26 ± 0.34 | 85.10 ± 1.04 | 73.18 ± 0.77 |
| GIK (Orsini et al., 2015) | **71.70 ± 0.79** | **76.31 ± 0.33** | 76.88 ± 0.47 | 83.07 ± 1.10 | **95.75 ± 0.50** |
| P2K (Neumann et al., 2016) | 69.22 ± 0.34 | – | 73.45 ± 0.48 | 91.70 ± 0.86 | 50.15 ± 1.92 |
| WWL (Togninalli et al., 2019) | **73.25 ± 0.87** | – | **77.91 ± 0.80** | – | – |

and 4 social network datasets (COLLAB, IMDB-BINARY, IMDB-MULTI, and REDDIT-BINARY) (Yanardag & Vishwanathan, 2015). For bioinformatics datasets, the categorical node labels are encoded as one-hot input features; for social network datasets, because nodes have no given features, we initialize all node features to 1. The 5 attributed graph datasets contain 3 bioinformatics datasets (ENZYMES, FRANKENSTEIN, PROTEINSatt) and 2 synthetic datasets (SYNTHETICNEW, Synthie). More detailed information can be found in Appendix A.4 Table 6.

**Baselines.** We compared our model with a number of state-of-the-art methods listed in the first column in Table 1 and 2 for simple graph classification and attributed graph classification, respectively. Besides GIN which inspired this work and the popular GCN (Kipf & Welling, 2017), we also include the recent studies on expressive power of GNNs, e.g., Ring-GNN (Chen et al., 2019), GHC (Nguyen & Maehara, 2020) and state-of-the-art methods on neighborhood aggregation, e.g., GraphSAGE (Hamilton et al., 2017) and HaarPool (Wang et al., 2020). For attributed graphs, few results on the benchmarks with deep learning methods are available in the literature. We are only aware of graph kernel related baselines, listed in the first columns in Table 2. We also compared iceGNN with our implemented GIN-final and GCN for attributed graph classification, both implemented by adjusting the corresponding official code.

**Results.** Table 1 and 2 list the classification accuracies on test set for simple graph classification and attributed graph classification, respectively. We highlight the top 3 accuracies for each dataset in boldface. From Table 1, for simple graph classification on bioinformatics datasets, iceGNN can achieve top 3 on all the 3 datasets except for NCI dataset where we achieve the best results among the deep learning models though. It seems that the graph kernel methods perform well on NCI1

Figure 2: The accuracy curves on training set in the training process.

datset. Especially, on the PTC dataset we achieve the best two performances, and improved 2 points compared to the past best. Although iceGNN can only achieve top 3 on REDDIT-BINARY dataset among the social network datasets, it can provide acceptable results overall. From Table 2, for all the attributed graph datasets, iceGNN can achieve top 3 in these 10 models, especially, iceGNN-MLP places first on 3 datasets. Comparing iceGNN and GIN-final, iceGNN can consistently outperform GIN-final except for iceGNN-fixed on ENZYMES dataset.

**Performance on training set.** To evaluate the expressive power, Figure 2(a-c) illustrates accuracies in training sets in the training process on 3 datasets for graph classification. More results on other datasets can be found in Appendix A.6 Figure 4. We can see that iceGNN-MLP on different datasets is able to fit the training sets perfectly and is better than GIN-final and iceGNN-fixed, specifically, iceGNN-MLP>iceGNN-fixed>GIN-final>GCN, in terms of the expressive power. For MUTAG and PTC datasets, GIN in Xu et al. (2019) can fit the training set well, while GIN-final cannot, since GIN in Xu et al. (2019) concatenates all the middle layer outputs as the graph embedding, it may be the reason that the final layer outputs of GIN may lose some information from middle layers.

We are aware of that iceGNN has more parameters than GIN-final (Appendix A.7 Table 8). To identify whether the expressive power is just due to more parameters, we enlarge GIN with 5 hidden layers in each MLP, so that the number of parameters achieve the same scale with iceGNN, denoting the GIN architecture as GIN-mlp5. We found that GIN-mlp5 still cannot achieve the training accuracy as iceGNN or even worse than GIN-final, as shown in Figure 2 (a-c).

**Recent GNN benchmarks.** The recent work by Dwivedi et al. (2020) proposed new GNN benchmarks by which we also test our model on their ZINC and MNIST datasets for graph regression and classification, respectively. To ensure a fair comparison, we followed their problem setting (data splits, optimizer, etc.) and GNN structure (number of layers, normalization). Our results together with some state-of-the-art results from Dwivedi et al. (2020) are listed in Table 3. From the results, we can see that iceGNN-MLP performs best on both ZINC and MNIST datasets among all the models, demonstrating its effectiveness.

Table 3: GNN performance on ZINC dataset for graph regression and on MNIST dataset for graph classification. Top 3 performances on each dataset are bolded. The best performances are underlined.

|  | ZINC | | MNIST | |
| --- | --- | --- | --- | --- |
|  | #Param | Test MAE | #Param | Test Acc |
| iceGNN-MLP | 231049 | **0.250±0.005** | 229140 | **97.340±0.083** |
| iceGNN-fixed | 252345 | 0.367±0.010 | 241568 | 96.490±0.124 |
| GIN | 103079 | 0.387±0.015 | 105434 | 96.485±0.252 |
| GCN | 505079 | 0.367±0.011 | 101365 | 90.705±0.218 |
| GatedGCN | 105735 | 0.435±0.011 | 104217 | **97.340±0.143** |
| GraphSage | 505341 | 0.398±0.002 | 104337 | **97.312±0.097** |
| MoNet | 504013 | **0.292±0.006** | 104049 | 90.805±0.032 |
| GAT | 531345 | 0.384±0.007 | 110400 | 95.535±0.205 |
| RingGNN | 97978 | 0.512±0.023 | 505182 | 91.860±0.449 |
| 3WLGNN | 102150 | 0.407±0.028 | 108024 | 95.075±0.961 |
| PNA | – | **0.320±0.032** | – | 97.190±0.080 |

Table 4: Node classification results. GCNII* is a variant of GCNII from Chen et al. (2020).

|  | Cora | Citeseer | Pubmed |
| --- | --- | --- | --- |
|  | Training accuracy | | |
| ICGNN-MLP | **1.000±0.000** | **0.999±0.000** | **0.975±0.002** |
| ICGNN-fixed | 0.998±0.004 | 0.990±0.011 | 0.904±0.053 |
| GIN-final | 0.998±0.001 | 0.974±0.002 | 0.935±0.004 |
| GCN | 0.977±0.002 | 0.946±0.003 | 0.841±0.003 |
| GCNII | 0.675±0.009 | 0.596±0.011 | 0.844±0.003 |
| GCNII* | 0.679±0.012 | 0.609±0.010 | 0.852±0.004 |
|  | Test accuracy | | |
| ICGNN-MLP | 0.858±0.019 | 0.728±0.023 | **0.888±0.008** |
| ICGNN-fixed | 0.841±0.025 | 0.720±0.025 | 0.872±0.027 |
| GIN-final | 0.870±0.018 | 0.739±0.031 | 0.875±0.009 |
| GCN | **0.883±0.014** | **0.773±0.020** | 0.859±0.007 |
| GCNII | 0.855±0.006 | 0.728±0.006 | 0.869±0.002 |
| GCNII* | 0.852±0.008 | 0.736±0.004 | 0.880±0.003 |

## 5.2 NODE CLASSIFICATION

We use three popular citation network datasets Cora, Citeseer, and Pubmed (Sen et al., 2008) for semi-supervised node classification. The detailed information of the dataset is summarized in Appendix A.4 Table 7. We compared our performance with a recent state-of-the-art, GCNII (Chen et al., 2020). Since the expressive power describes the ability of a model to discriminate different nodes, a larger training set can reflect the expressive power better. Because the official splits have only a few nodes in training, we split the nodes into training, validation and test sets by 8:1:1. The node classification results on test sets are listed in Table 4, where we found that iceGNNs and can outperform all other baselines on training set; especially on Cora and Citeseer dataset, iceGNNs achieve nearly 100% accuracy, suggesting iceGNNs have strong expressive ability. For test set, iceGNN-MLP performs better than all other models on Pubmed dataset, but does not perform that well on Cora and Citeseer datasets comparing to GCN and GIN-final, suggesting that iceGNN-MLP also has good generalization ability on Pubmed dataset rather than Cora and Citeseer datasets.

## 5.3 EXPRESSIVE CAPABILITY ANALYSIS

The expressive power describes how a model can distinguish different samples. Generally, a highly expressive model will map different samples to different embeddings and similar samples to similar embeddings. For classification problem, an expressive model should make samples in the same class compact together and samples in different classes highly discriminative. Here, we fetch the output embeddings of GNN before feeding to the classifier, and visualize them to see if GNN can discriminate samples from different classes. Figure 3 shows the t-SNE visualization (Maaten & Hinton, 2008) of output graph representations of different GNN models on training data of NCI1 dataset. More visualization results can be found in Appendix A.8. We can see that the output embeddings of iceGNN-MLP and iceGNN-fixed are discriminative on both datasets, and the less expressive GIN-final shows somewhat more overlaps between different classes, which validates the expressive capability of iceGNN.

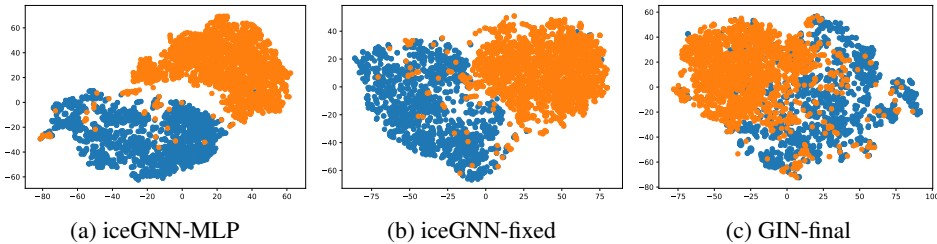

| (a) iceGNN-MLP | (b) iceGNN-fixed | (c) GIN-final |

Figure 3: t-SNE visualization of the output embeddings on training data of NCI1 dataset.

From the experimental results, we found that iceGNN-MLP often performs better than iceGNN-fixed. We identify two reasons could make iceGNN-MLP outperform iceGNN-fixed. (1) By the discussion in Section 4.2, iceGNN-MLP can also retain the graph label information. (2) iceGNN-fixed usually has a much higher dimension input to each GNN layer, thus more parameters to train, having a high risk of encountering local optimum and plateau. This phenomenon also exists in general MLP.

## 6 CONCLUSION

In this paper, we present a theoretical framework to design highly expressive GNNs for general graphs. Based on the framework, we propose two iceGNN variants with fixed transformation function and learnable transformation function, respectively. Moreover, the proposed iceGNN can naturally learn expressive representations for graphs with continuous node attributes. We validate the proposed GNN for graph classification and node classification on multiple benchmark datasets, including simple graphs and attributed graphs. The experimental results demonstrate that our model achieves state-of-the-art performances on most of the benchmarks. Future directions include extending the framework to graph with continuous edge attributes.

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

# A APPENDIX

## A.1 COMMON NOTATIONS USED THROUGHOUT THE PAPER

Table 5: Common notations used throughout the paper.

| Notation | Definition |
|---|---|
| $\mathbb{R}$ | the set of all real numbers |
| $\mathbb{N}$ | the set of all natural numbers |
| $[\mathbf{x}_1, \cdots, \mathbf{x}_n]$ | the concatenation of vectors $\mathbf{x}_1, \cdots, \mathbf{x}_n$, |
| $H^{(k)}(v)$ | the vector representation of node $v$ in the $k$th layer |
| $\mathcal{N}(v)$ | the neighborhood of node $v$ in a graph |
| $\mathbf{x}[i]$ | the $i$th entry of vector $\mathbf{x}$ |
| $\mathbf{x}[i:j]$ | the subvector of $\mathbf{x}$ between index $i$ and $j$ (including) |

A.2 PROOF OF THEOREM 2

**Theorem 2** *Assume $\mathcal{X}_M$ is a set of finite subsets of $\mathbb{R}^d$ with size less than or equal to $M$, i.e., for $M \in \mathbb{N}$ and $\mathcal{X}_M = \{X | X \subset \mathbb{R}^d, |X| \leq M\}$, there exists an infinite number of continuous functions $\Phi : \mathbb{R}^d \to \mathbb{R}^D$ such that the set function $f : \mathcal{X}_M \to \mathbb{R}^D$, $f(X) = \sum_{\mathbf{x} \in X} \Phi(\mathbf{x})$ is continuous and injective.*

*Proof.* We prove the theorem by three steps, 1. constructing a satisfying function $\Phi(x)$ in one dimensional cases ($d = 1$); 2. constructing a satisfying function $\Phi(\mathbf{x})$ in multi-dimensional cases ($d > 1$); 3. The number of the satisfying functions is infinite.

**1. One dimensional cases ($d = 1$):**

In one dimensional case, the theorem can be easily proved by extending the following lemma from Zaheer et al. (2017).

**Lemma.** *Let $\mathcal{X} = \{(x_1, \cdots, x_M) \in [0, 1]^M : x_1 \leq x_2 \leq \cdots \leq x_M\}$. The sum-of-power mapping $E : \mathcal{X} \to \mathbb{R}^{M+1}$ defined by the coordinate functions*

$$E(X) = [E_0(X), E_1(X), \cdots, E_M(X)] = \left[ \sum_{x \in X} 1, \sum_{x \in X} x, \cdots, \sum_{x \in X} x^M \right] \quad (12)$$

*is injective.*

In Zaheer et al. (2017), this lemma is proved based on the famous Newton-Girard formulae, where the domain $\mathcal{X}$ can be extended to $\mathcal{X}_M = \{X | X \subset \mathbb{R}, |X| \leq M, M \in \mathbb{N}\}$ with the same proof process. Because $E_0(X) = \sum_{x \in X} 1 = |X|$ is the number of elements in $X$, $E_0(X_1) = E_0(X_2)$ implies equal set size between the two sets, it can be easily extended to $\mathcal{X}_M = \{X | X \subset \mathbb{R}, |X| \leq M, M \in \mathbb{N}\}$.

Since $E(X) = \left[\sum_{x \in X} 1, \sum_{x \in X} x, \cdots, \sum_{x \in X} x^M\right] = \sum_{x \in X} [1, x, \cdots, x^M]$. Let $\Phi(x) = [1, x, \cdots, x^M]$, obviously $\Phi(x)$ is continuous, thus, we get one $\Phi(X)$ such that $f(X) = \sum_{x \in X} \Phi(x)$ is injective and continuous.

**2. Multi-dimensional cases ($d > 1$):**

For a $d$-dimensional vector $\mathbf{x}$ and a integer $M$, we define

$$P_{i,j}(\mathbf{x}; M) = \left[\mathbf{x}[j], \mathbf{x}[i]\mathbf{x}[j], \mathbf{x}[i]^2\mathbf{x}[j], \cdots, \mathbf{x}[i]^{M-1}\mathbf{x}[j]\right] \quad (13)$$

where $i, j \in [1, d]$, $\mathbf{x}[i]$ is the $i$th entry of $\mathbf{x}$.

For a given $i \in [1, d]$, define $\Phi_i : \mathbb{R}^d \to \mathbb{R}^{(dM+1)}$, $\Phi_i(\mathbf{x}; M) = [1, P_{i,1}, \cdots, P_{1,d}]$, we will prove that $\Phi_i(\mathbf{x}; M)$ can meet the condition that $f(X) = \sum_{\mathbf{x} \in X} \Phi_i(\mathbf{x}; M)$ is injective and continuous if $\mathbf{x}[i]$ is distinct for $\mathbf{x} \in X$.

We only consider $\Phi_1(\mathbf{x}; M)$, for $i = 2, \cdots, d$, the situations are similar. For a clear understanding, we reshape $\Phi_1(\mathbf{x}; M)$ to $d$ rows similar to a matrix, like

$$\Phi_1(\mathbf{x}; M) = \begin{bmatrix} 1, \mathbf{x}[1], & \mathbf{x}[1]^2, \cdots, & \mathbf{x}[1]^M, \\ \mathbf{x}[2], & \mathbf{x}[1]\mathbf{x}[2], \cdots, & \mathbf{x}[1]^{M-1}\mathbf{x}[2], \\ \vdots & \vdots \ddots & \vdots \\ \mathbf{x}[d], & \mathbf{x}[1]\mathbf{x}[d], \cdots, & \mathbf{x}[1]^{M-1}\mathbf{x}[d] \end{bmatrix} \quad (14)$$

For a $(dm + 1)$-dimensional vector $V$ from the image domain of $\mathcal{X}_M$ through $f(X) = \sum_{\mathbf{x} \in X} \Phi_1(\mathbf{x}; M)$, we will identify the number of the preimages of $V$.

Let $X$ is a preimage of $V$, we have the following equation which is exactly a equation group with $dM + 1$ equations.

$$V = \begin{bmatrix} \sum_{\mathbf{x} \in X} 1, & \sum_{\mathbf{x} \in X} \mathbf{x}[1], & \sum_{\mathbf{x} \in X} \mathbf{x}[1]^2, & \cdots, & \sum_{\mathbf{x} \in X} \mathbf{x}[1]^M, \\ & \sum_{\mathbf{x} \in X} \mathbf{x}[2], & \sum_{\mathbf{x} \in X} \mathbf{x}[1]\mathbf{x}[2], & \cdots, & \sum_{\mathbf{x} \in X} \mathbf{x}[1]^{M-1}\mathbf{x}[2], \\ & \vdots & \vdots & \ddots & \vdots \\ & \sum_{\mathbf{x} \in X} \mathbf{x}[d], & \sum_{\mathbf{x} \in X} \mathbf{x}[1]\mathbf{x}[d], & \cdots, & \sum_{\mathbf{x} \in X} \mathbf{x}[1]^{M-1}\mathbf{x}[d] \end{bmatrix} \quad (15)$$

Note that the first row of Eq. 15 is exactly the sum-of-power mapping we considered in one dimensional cases, thus we can identify a unique set of the first entry of elements in $X$, and the number of elements in $X$ is also determined. Let $X$ have $M$ elements and $X = \{\mathbf{x_1}, \mathbf{x_2}, \cdots, \mathbf{x_M}\}$, the set $\{\mathbf{x_1}[1], \mathbf{x_2}[1], \cdots, \mathbf{x_M}[1]\}$ is uniquely defined.

Consider the second row of Eq. 15, we can rewrite the equations in the second row as linear matrix equation as Eq. 16

$$
\begin{bmatrix}
1 & 1 & \cdots & 1 \\
\mathbf{x_1}[1] & \mathbf{x_2}[1] & \cdots & \mathbf{x_M}[1] \\
\mathbf{x_1}[1]^2 & \mathbf{x_2}[1]^2 & \cdots & \mathbf{x_M}[1]^2 \\
\vdots & \vdots & \ddots & \vdots \\
\mathbf{x_1}[1]^{(M-1)} & \mathbf{x_2}[1]^{(M-1)} & \cdots & \mathbf{x_M}[1]^{(M-1)}
\end{bmatrix}
\times
\begin{bmatrix}
\mathbf{x_1}[2] \\
\mathbf{x_2}[2] \\
\vdots \\
\mathbf{x_M}[2]
\end{bmatrix}
= V[2,:]^T
\tag{16}
$$

Note that the coefficient matrix in left side of Eq. 16 is a Vandermonde matrix, if the $\mathbf{x_1}[1], \cdots, \mathbf{x_M}[1]$ are all distinct, the coefficient matrix is invertible (Macon & Spitzbart, 1958), Eq. 16 has a unique solution for $\mathbf{x_1}[2], \cdots, \mathbf{x_M}[2]$ corresponding to $\mathbf{x_1}[1], \cdots, \mathbf{x_M}[1]$. Similarly, by the $i$th $(2 < i < d)$ row of Eq. 15, $\mathbf{x_1}[i], \cdots, \mathbf{x_M}[i]$ can be uniquely identified.

In the other case, if the $\mathbf{x_1}[1], \cdots, \mathbf{x_M}[1]$ that solved from the first row of Eq. 15 are not all distinct, Eq. 16 has infinitely many solutions. $\Phi(\mathbf{x})$ defined by Eq. 14 is not sufficient to make $f(X) = \sum_{\mathbf{x} \in X} \Phi(\mathbf{x})$ injective. We need some more dimensions appended in $\Phi(\mathbf{x})$.

Let $\mathbf{x_1}[1] = \mathbf{x_2}[1] = \cdots = \mathbf{x_k}[1]$, then by combining the items, Eq. 16 is shrinked to

$$
\begin{bmatrix}
1 & 1 & \cdots & 1 \\
\mathbf{x_k}[1] & \mathbf{x_{k+1}}[1] & \cdots & \mathbf{x_M}[1] \\
\mathbf{x_k}[1]^2 & \mathbf{x_{k+1}}[1]^2 & \cdots & \mathbf{x_M}[1]^2 \\
\vdots & \vdots & \ddots & \vdots \\
\mathbf{x_k}[1]^{(M-1)} & \mathbf{x_{k+1}}[1]^{(M-1)} & \cdots & \mathbf{x_M}[1]^{(M-1)}
\end{bmatrix}
\times
\begin{bmatrix}
\sum_{i=1 \cdots k} \mathbf{x_i}[2] \\
\mathbf{x_{k+1}}[2] \\
\vdots \\
\mathbf{x_M}[2]
\end{bmatrix}
= V[2,:]^T
\tag{17}
$$

By solving Eq. 17, we have a unique sum $\sum_{i=1 \cdots k} \mathbf{x_i}[2]$. To identify a unique set of $\{\mathbf{x_1}[2], \mathbf{x_2}[2], \cdots, \mathbf{x_k}[2]\}$, we can define a unique $\sum_{i=1 \cdots k} \mathbf{x_i}[2]^2, \sum_{i=1 \cdots k} \mathbf{x_i}[2]^3, \cdots, \sum_{i=1 \cdots k} \mathbf{x_i}[2]^k$, we can add items $\mathbf{x}[2]^2, \mathbf{x}[1]\mathbf{x}[2]^2, \mathbf{x}[1]^2\mathbf{x}[2]^2, \cdots, \mathbf{x}[1]^{M-1}\mathbf{x}[2]^2$ to $\Phi(\mathbf{x})$ to uniquely identify $\sum_{i=1 \cdots k} \mathbf{x_i}[2]^2$. Similarly, add items $\mathbf{x}[2]^k, \mathbf{x}[1]\mathbf{x}[2]^k, \mathbf{x}[1]^2\mathbf{x}[2]^k, \cdots, \mathbf{x}[1]^{M-1}\mathbf{x}[2]^k$ to $\Phi(\mathbf{x})$ to uniquely identify $\sum_{i=1 \cdots k} \mathbf{x_i}[2]^k$. Thus all $\mathbf{x_i}[2]$ are identified.

After the set $\{\mathbf{x_1}[2], \mathbf{x_2}[2], \cdots, \mathbf{x_M}[2]\}$ is uniquely defined, by adding $\mathbf{x}[2]^i\mathbf{x}[j] (i = 0, \cdots M - 1, j = 3, \cdots d)$ to $\Phi(\mathbf{x})$, we can use $\mathbf{x_i}[2]$ to construct a Vandermonde matrix to solve $\mathbf{x_i}[3], \cdots, \mathbf{x_i}[d], (i = 1, \cdots, M)$. If $\mathbf{x_i}[1:2]$ are not all distinct, we can identify $\mathbf{x_i}[3]$ similarly by adding $\mathbf{x}[1]^i\mathbf{x}[3]^j$ to $\Phi(\mathbf{x})$. By this way, the set $X$ can be uniquely identified.

In our construction of $\Phi(\mathbf{x})$, all the functions are continuous, thus, there exists a continuous function $\Phi(\mathbf{x})$ such that $f(X) = \sum_{\mathbf{x} \in X} \Phi(\mathbf{x})$ is injective and continuous.

**3. The number of the satisfying function is infinity:**

To prove the number of this kind functions is infinity, we construct a continuous injective function $g : \mathbb{R} \to \mathbb{R}$, we will show that if we have a $\phi(\mathbf{x})$ satisfying the condition $f(X) = \sum_{\mathbf{x} \in X} \phi(\mathbf{x})$ is continuous and injective, then $\phi(g(\mathbf{x}))$ also satisfy the condition $f(X) = \sum_{\mathbf{x} \in X} \phi(g(\mathbf{x}))$ is continuous and injective.

We define a function $h : \mathcal{X} \to \mathcal{X}$, $h(X) = \{g(\mathbf{x})|\mathbf{x} \in X\}$, since $g(\mathbf{x})$ is injective, $h(X)$ is also injective. If we have a function $\phi(\mathbf{x})$ such that $f(X) = \sum_{\mathbf{x} \in X} \phi(\mathbf{x})$ is injective, $f(h(X))$ is injective.

$$
f(h(X)) = \sum_{\mathbf{x} \in h(X)} \phi(\mathbf{x}) = \sum_{\mathbf{x} \in \{g(\mathbf{x})|\mathbf{x} \in X\}} \phi(\mathbf{x}) = \sum_{\mathbf{x} \in X} \phi(g(\mathbf{x}))
\tag{18}
$$

Because $\phi(\mathbf{x})$ and $g(\mathbf{x})$ are both continuous, $\phi(g(\mathbf{x}))$ is continuous, thus we find another function $\phi(g(\mathbf{x}))$ such that $\sum_{\mathbf{x} \in X} \phi(g(\mathbf{x}))$ is injective. Because we can have an infinite number of such continuous injective functions $g : \mathbb{R}^d \to \mathbb{R}^d$ (e.g., $g(\mathbf{x}) = k\mathbf{x}, k \in \mathbb{R}$), we have a infinite number of such functions $\Phi(\mathbf{x}) = \phi(g(\mathbf{x}))$ such that $f(X) = \sum_{\mathbf{x} \in X} \Phi(\mathbf{x})$ is injective.

$\square$

### A.3 PROOF OF THEOREM 3

**Theorem 3** *Let $M \in \mathbb{N}$ and $\mathcal{X} = \{X | X \subset \mathbb{R}^d, |X| = M\}$, then for any continuous function $\Phi : \mathbb{R}^d \to \mathbb{R}^N$, if $N < dM$, the set function $f : \mathcal{X} \to \mathbb{R}^N$ $f(X) = \sum_{\mathbf{x} \in X} \Phi(\mathbf{x})$ is not injective.*

*Proof.* Suppose $f(X)$ is injective. Because $\Phi(\mathbf{x})$ is continuous, $f(X)$ is a finite sum of continuous function, it is also continuous, thus, $f(X)$ is continuous and injective.

All sets in $\mathcal{X}$ have $M$ elements from $\mathbb{R}^d$. In one dimensional cases ($d = 1$), $\mathcal{X}$ has a bijection to $\mathcal{S} = \{X = (x_1, \cdots, x_M) | X \in \mathbb{R}^M, x_1 \le x_2 \le \cdots \le x_M\}$.

In multi-dimensional cases ($d > 1$), we can construct a bijection from $\mathcal{X}$ to $\mathcal{S} = \{X = (\mathbf{x_1}, \cdots, \mathbf{x_M}) | X \in \mathbb{R}^{dM}, \mathbf{x_1}[1] \le \mathbf{x_2}[1] \le \cdots \le \mathbf{x_M}[1], \text{if } \mathbf{x_i}[1 : k] = \mathbf{x_{i+1}}[1 : k], \mathbf{x_i}[k + 1] \le \mathbf{x_{i+1}}[k + 1], i = 1, \cdots M - 1, k = 1, \cdots, d - 1\}$. For $X \in \mathcal{X}$, let $X = \{\mathbf{x_1}, \cdots, \mathbf{x_M}\}$, we can sort elements in $X$ by the first entry, for the elements whose first entries are equal, sort them by the second entry, so repeatedly in this way, we get a final ordered sequence of the vectors, which is unique in $\mathcal{S}$.

Note that $\mathcal{S}$ is a convex open subset of $\mathbb{R}^{dM}$, and is therefore homeomorphic to $\mathbb{R}^{dM}$. Since $N < dM$, no continuous injection exists from $\mathbb{R}^{dM}$ to $\mathbb{R}^N$. Thus no continuous injective function exist from $\mathcal{X}$ to $\mathbb{R}^N$. Hence we have reached a contradiction. $\square$

## A.4 DATASET DETAILS

The dataset information is in Table 6 and 7

Table 6: Dataset information for graph classification. All datasets are available from Kersting et al. (2016). #G=number of graphs. #C=number of classes. #NC=number of node types. AvgN=average number of nodes in one graph. AvgE=average number of edges in one graph. Dim=node attribute dimension. MaxNb is the max 1-hop neighbors in all the nodes.

| | #G | #C | #NC | AvgN | AvgE | MaxNb | Dim | Type | Source |
|---|---|---|---|---|---|---|---|---|---|
| MUTAG | 188 | 2 | 7 | 17.93 | 19.79 | 4 | – | bioinformatics | (Debnath et al., 1991) |
| PTC | 344 | 2 | 19 | 14.29 | 14.69 | 4 | – | bioinformatics | (Helma et al., 2001) |
| NCI1 | 4110 | 2 | 37 | 29.87 | 32.30 | 4 | – | bioinformatics | (Wale et al., 2008) |
| PROTEINS | 1113 | 2 | 3 | 39.06 | 72.82 | 25 | – | bioinformatics | (Borgwardt et al., 2005) |
| COLLAB | 5000 | 3 | 1 | 74.49 | 2457.78 | 491 | – | social networks | (Yanardag & Vishwanathan, 2015) |
| IMDB-B | 1000 | 2 | 1 | 19.77 | 96.53 | 135 | – | social networks | (Yanardag & Vishwanathan, 2015) |
| IMDB-M | 1500 | 3 | 1 | 13.00 | 65.94 | 88 | – | social networks | (Yanardag & Vishwanathan, 2015) |
| RDT-B | 2000 | 2 | 1 | 429.63 | 497.75 | 3062 | – | social networks | (Yanardag & Vishwanathan, 2015) |
| ENZYMES | 600 | 6 | 3 | 32.63 | 62.14 | 9 | 18 | bioinformatics | (Borgwardt et al., 2005) |
| FRANKENSTEIN | 4337 | 2 | 1 | 16.90 | 17.88 | 4 | 780 | bioinformatics | (Orsini et al., 2015) |
| PROTEINS-att | 1113 | 2 | 3 | 39.06 | 72.82 | 25 | 1 | bioinformatics | (Borgwardt et al., 2005) |
| SYNTHETICnew | 300 | 2 | 1 | 100.00 | 196.25 | 9 | 1 | synthetic | (Feragen et al., 2013) |
| Synthie | 400 | 4 | 1 | 95.00 | 172.93 | 20 | 15 | synthetic | (Morris et al., 2016) |

Table 7: Dataset information for node classification.

| | #Nodes | #Edges | #Classes | Dim | MaxNb | Source |
|---|---|---|---|---|---|---|
| cora | 2708 | 5429 | 7 | 1433 | 168 | (Sen et al., 2008; Kipf & Welling, 2017) |
| citeseer | 3327 | 4732 | 6 | 3703 | 99 | (Sen et al., 2008; Kipf & Welling, 2017) |
| pubmed | 19717 | 44338 | 3 | 500 | 171 | (Sen et al., 2008; Kipf & Welling, 2017) |

## A.5 IMPLEMENTATION DETAILS

We implement 2 iceGNN variants: (1) iceGNN-fixed, where fixed transformation functions in all layers are set as $\Phi_M(\mathbf{x})$ or $\Phi'_M(\mathbf{x})$ in Eq. (4) in the main paper; (2) iceGNN-MLP, where the transformation functions in all layers are set as a learnable MLP. Since for simple graph with one-hot node features, the summation with identical transformation is injective, we set whether the transformation function in the first layer is identical or an MLP as an optional hyperparameter for simple graph classification. For social network datasets, the max neighborhood size is too large that a fixed transformation function will produce a large hidden dimension, we do not implement fixed transformation function. Also, because the nodes have no initial features in the first layer. We also implement GIN with the output of the final layer as node embeddings to sum to graph embedding, denoted as GIN-final.

The two iceGNN variants and GIN-final are implemented with 5 layers, all MLPs in iceGNN have 2 layers. Batch normalization Ioffe & Szegedy (2015) is applied in every hidden layer (including GNN layer and MLP layer) followed by a ReLU activation function. We use the Adam optimizer Kingma & Ba (2015) with a initial learning rate and decay the learning rate by 0.5 every 50 epochs. The batch size is 32, no dropout layer applied. The search space of hyper-parameters we tuned for each dataset are: (1) The number of hidden units $\{16, 32, 64\}$; (2) the inital learning rate $0.01, 0.001$; (3) for the 4 bioinformatic simple graph datasets, the transformation function in the first layer is set identical or MLP; (4) For iceGNN-fixed, the transformation function is tuned with $\Phi_M(\mathbf{x})$ or $\Phi'_M(\mathbf{x})$ in Eq. (4) in the main paper. For each dataset, we follow the standard 10-fold cross validation protocol and use the same splits with Xu et al. (2019). Following the previous work Xu et al. (2019); Maron et al. (2019a); Mao et al. (2020), we reported the best averaged validation accuracy across the 10 folds for a fair comparison. All models are trained 300 epochs. To evaluate the expressive capability, we also record the average training accuracy across the 10 folds of iceGNNs and GIN-final in each epoch. All the experiments were run in 10 Tesla V100 GPUs with pytorch.

### A.6 ADDITIONAL RESULTS

The accuracy curves on training set in the training process on more datasets are shown in Figure 4, where Figures 4a-4j are for graph classification and Figures 4k-4l are for node classification.

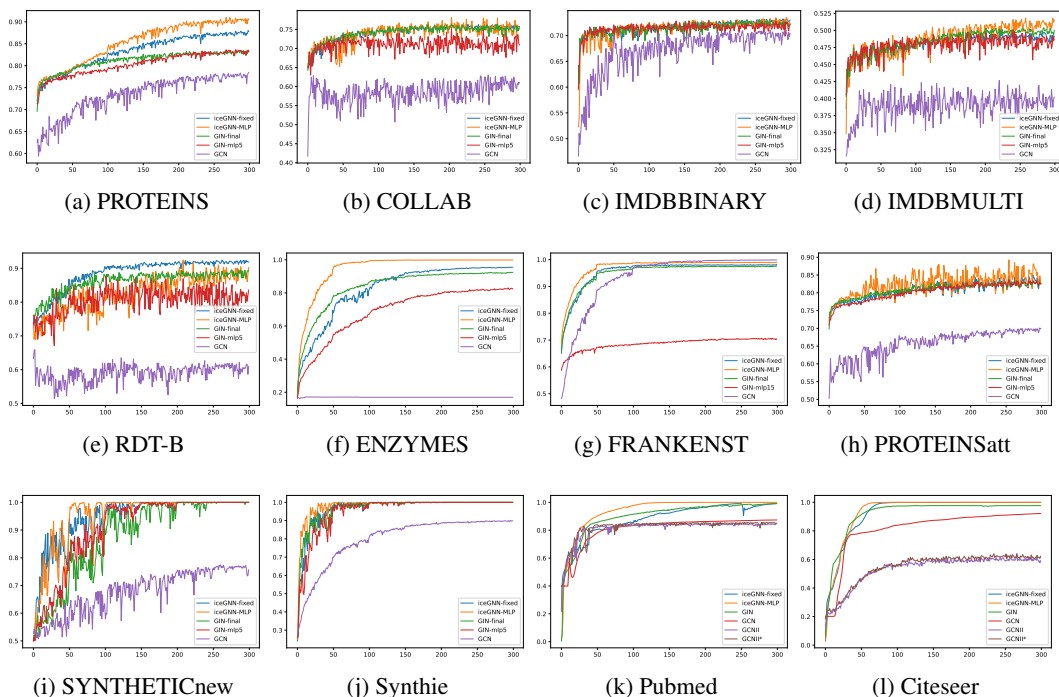

Figure 4: The accuracy curves on training set in the training process. All the models have a hidden dimension 16.

A.7    MODEL SIZE

Table 8 lists the comparison of number of trainable parameters in different models on different datasets. GIN-mlp(n), the numbers in parentheses for different datasets are the number of layers in an MLP. GIN-mlp(n) is implemented to enlarge the GIN model to the same parameter scale with iceGNN-MLP to make a fair comparison of expressive power. All the models have a hidden dimension 16, GIN-mlp(n) is a GIN model where all MLPs are implemented with n layers. Since iceGNN-fixed cannot be implemented with a large MaxNb (MaxNb is defined in Tables 6 and 7), the transformation functions in iceGNN-fixed are set as $\Phi_4(\mathbf{x})$ (defined in Eq. (5) in the main paper). The results show that, iceGNN is more expressive than GIN on most of the datasets, and the expressive power of iceGNN comes from the injective and continuous aggregation scheme rather than the number of parameters.

Table 8: Number of trainable parameters of different models on different datasets. All the models have a hidden dimension 16.

| models | iceGNN-fixed | iceGNN-MLP | GIN-final | GIN-mlp(n) | GCN |
|---|---|---|---|---|---|
| MUTAG | 8,706 | 8,574 | 4,256 | 8,816(5) | 978 |
| PTC | 9,666 | 9,006 | 4,472 | 9,032(5) | 1,170 |
| NCI1 | 11,106 | 9,654 | 4,796 | 9,356(5) | 1,458 |
| PROTEINS | 8,226 | 8,430 | 4,184 | 8,744(5) | 914 |
| COLLAB | 8,227 | 7,815 | 4,213 | 8,773(5) | 899 |
| IMDBBINARY | 8,162 | 7,750 | 4,148 | 8,708(5) | 882 |
| IMDBMULTI | 8,227 | 7,815 | 4,213 | 8,773(5) | 899 |
| REDDITBINARY | 8,162 | 7,750 | 4,148 | 8,708(5) | 882 |
| ENZYMES | 10,086 | 9,338 | 4,768 | 9,328(5) | 1,270 |
| FRANKENSTEIN | 70,546 | 36,402 | 18,170 | 37,930(15) | 13,346 |
| PROTEINSatt | 34,892 | 8,466 | 4,202 | 8,762(5) | 930 |
| Synthie | 9,476 | 8,992 | 4,530 | 9,090(5) | 1,140 |
| SYNTHETICnew | 8,226 | 8,358 | 4,148 | 8,708(5) | 882 |

## A.8 VISUALIZATION

The graph embedding output from the final layer of GNNs on some other datasets is visualized with t-SNE (Figures 5,6,7,8).

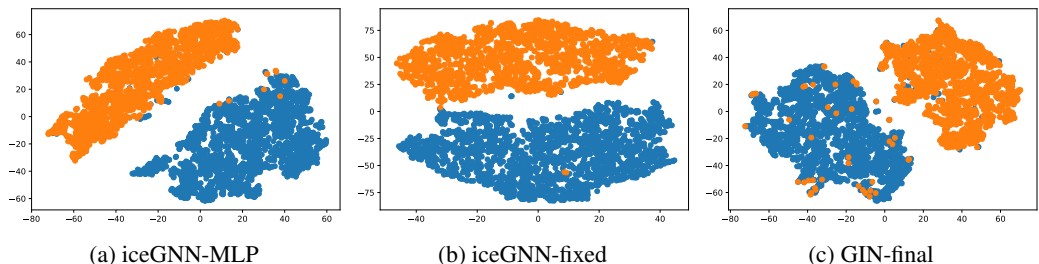

(a) iceGNN-MLP          (b) iceGNN-fixed          (c) GIN-final

Figure 5: t-SNE visualization of the output 16-dimensional graph embeddings on training data of FRANKENSTEIN dataset.

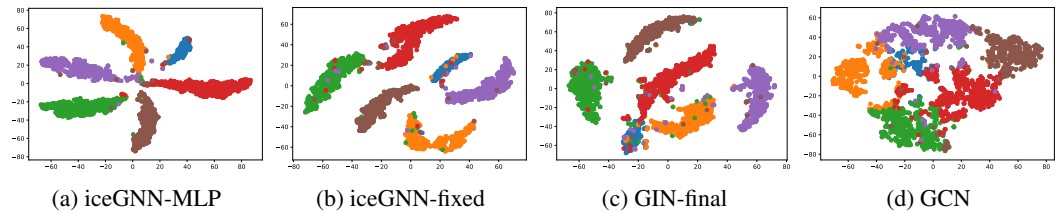

(a) iceGNN-MLP          (b) iceGNN-fixed          (c) GIN-final          (d) GCN

Figure 6: t-SNE visualization of the output 16-dimensional node embeddings on training data of Citeseer dataset.

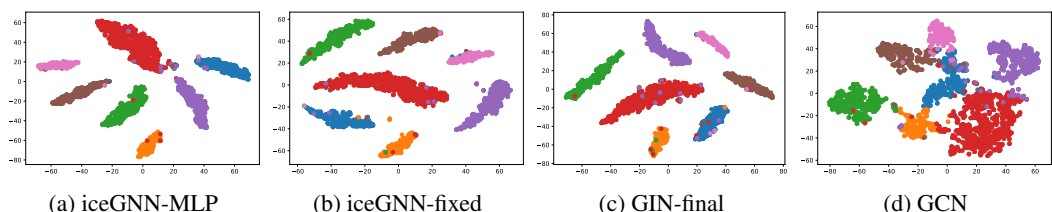

(a) iceGNN-MLP          (b) iceGNN-fixed          (c) GIN-final          (d) GCN

Figure 7: t-SNE visualization of the output 16-dimensional node embeddings on training data of Cora dataset.

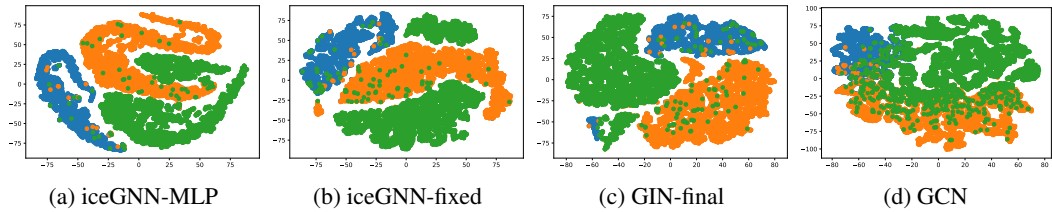

(a) iceGNN-MLP          (b) iceGNN-fixed          (c) GIN-final          (d) GCN

Figure 8: t-SNE visualization of the output 16-dimensional node embeddings on training data of Pubmed dataset.

### A.9 HYPER-PARAMETER SETTING

For the reproducibility, Table 9 provides the hyper-parameters to achieve the results of iceGNN in Table 1 and 2 in the main paper, and the code is also attached with this file. For the social network datasets, since MaxNB is too large to implement iceGNN-fixed, we only implement iceGNN-MLP. Since all nodes are of the same type, identical transformation function in the first layer is applied. For attributed graphs, to preserve injective and continuous, the transformation function in the first layer is not identical. The results of iceGNN, GIN and GCN in Table 4 in the main paper is achieved hyper-parameters in Table 10, and all the learning rates are 0.01. The hyper-parameters of GCNII and GCNII* are the same as the official code (`https://github.com/chennnM/GCNII`), where the hidden dimensions are 64, 256, 256 for Cora, Citeseer and Pubmed, respectively.

Table 9: The hyper-parameters corresponding to the results of iceGNN in Table 1 and 2 in the main paper. h=hidden dimension; lr=learning rate; TF=transformation function; FI=whether to apply a identical transformation function for the first layer.

| dataset | model | h | lr | FI | TF |
|---|---|---|---|---|---|
| MUTAG | iceGNN-fixed | 16 | 0.01 | FALSE | $\Phi$ |
| | iceGNN-MLP | 16 | 0.01 | FALSE | MLP |
| PTC | iceGNN-fixed | 16 | 0.001 | TRUE | $\Phi'$ |
| | iceGNN-MLP | 16 | 0.001 | FALSE | MLP |
| NCI1 | iceGNN-fixed | 64 | 0.001 | TRUE | $\Phi$ |
| | iceGNN-MLP | 16 | 0.001 | TRUE | MLP |
| PROTEINS | iceGNN-fixed | 64 | 0.001 | TRUE | $\Phi'$ |
| | iceGNN-MLP | 64 | 0.01 | TRUE | MLP |
| COLLAB | iceGNN-MLP | 64 | 0.001 | TRUE | MLP |
| IMDB-B | iceGNN-MLP | 16 | 0.01 | TRUE | MLP |
| IMDB-M | iceGNN-MLP | 16 | 0.01 | TRUE | MLP |
| RDT-B | iceGNN-MLP | 16 | 0.01 | TRUE | MLP |
| ENZYMES | iceGNN-fixed | 64 | 0.001 | FALSE | $\Phi$ |
| | iceGNN-MLP | 64 | 0.001 | FALSE | MLP |
| FRANKENSTEIN | iceGNN-fixed | 32 | 0.001 | FALSE | $\Phi'$ |
| | iceGNN-MLP | 16 | 0.001 | FALSE | MLP |
| PROTEINSatt | iceGNN-fixed | 16 | 0.001 | FALSE | $\Phi'$ |
| | iceGNN-MLP | 64 | 0.01 | FALSE | MLP |
| SYNTHETICnew | iceGNN-fixed | 64 | 0.01 | FALSE | $\Phi$ |
| | iceGNN-MLP | 64 | 0.01 | FALSE | MLP |
| Synthie | iceGNN-fixed | 64 | 0.01 | FALSE | $\Phi$ |
| | iceGNN-MLP | 32 | 0.01 | FALSE | MLP |

Table 10: The hyper-parameters corresponding to the results of iceGNN in Table 4 in the main paper. h=hidden dimension; TF=transformation function; FI=whether to apply a identical transformation function for the first layer.

| dataset | model | h | lr | FI | TF | drop out rate | weight decay |
|---|---|---|---|---|---|---|---|
| Cora | iceGNN-fixed | 128 | 0.01 | TRUE | $\Phi$ | 0 | 0.01 |
| | iceGNN-MLP | 128 | 0.01 | TRUE | MLP | 0.6 | 0 |
| Citesser | iceGNN-fixed | 128 | 0.01 | TRUE | $\Phi$ | 0 | 0.01 |
| | iceGNN-MLP | 128 | 0.01 | TRUE | MLP | 0.6 | 0 |
| Pubmed | iceGNN-fixed | 16 | 0.01 | TRUE | $\Phi$ | 0 | 0.01 |
| | iceGNN-MLP | 16 | 0.01 | TRUE | MLP | 0.6 | 0 |

