# OpenReview forum: "Towards Expressive Graph Representations for Graph Neural Networks"
_ICLR.cc/2023/Conference — Submitted to ICLR 2023_

### Official Review · Reviewer_EYLv · 2022-10-20

**Confidence:** 3
**Correctness:** 4
**Technical Novelty And Significance:** 2
**Empirical Novelty And Significance:** 2
**Recommendation:** 5

**Clarity, Quality, Novelty And Reproducibility:**

The paper is clearly written and the source code is provided in the appendix. Moreover, all datasets are publicly available, ensuring reproducibility.

I have some doubts about the novelty; see above. For example, the simple layer proposed by Morris et al. (2019) is already injective *and* continuous. The proof of Theorem 2 is rather straightforwardly based on known results.

**Strength And Weaknesses:**

**Strength**
- Paper is well presented and easy-to-read
- Good empirical results, good experimental study
- Theory seems, to some extent, align with experimental results

**Weaknesses**
- Not clear enough, if existing works already solve the problem of infectivity and continuity; see below
- Some weaknesses in the experimental protocol; see below

**Questions**
- Why is the function considered by Xu et al. (2019) not continuous? To the best of my knowledge, it is a composition of continuous functions.


**Suggestions/Remarks**:
- The simple layer proposed by Morris et al. (2019) is already injective *and* continuous.
- You assume a finite subset of $\mathbb{R}^d$. Hence, it is countable and can be handled by the approaches of Morris et al. (2019) or Xu et al. (2019)
- Make more clear that you deal with multiset functions and not set functions
- In the abstract, you claim "to improve the expressive power of GNN". However, existing works, e.g., Xu et al. (2019), are already maximal expressive.
- Page 2: Corso et al. (2020) does not increase the expressivity of GNNs
- In the experiments, it would beneficial to also include larger datasets from OGB or TUDataset with continuous node features. The used datasets are rather small
- The choice of reporting validations set performance is questionable even if other papers have done so in the past.

**Summary Of The Paper:**

The paper investigates GNNs' expressive power. That is, building on the work of Xu et al. (2019) and Morris et al. (2019) it proposes injective and continuous aggregating functions used within GNNs. Specifically, the paper argues that both injectiveness, as well as continuity, is needed to compute meaningful graph and node representations. To that, the paper proposes a simple injective and continuous function for multisets by building on results by Zaheer et al. (2017). Moreover, they derive sufficient conditions for injectivity.

The theoretical results are complemented with an empirical study, showcasing the benefits of the new aggregation function on a number of standard benchmark datasets.

**Summary Of The Review:**

This is a well-written paper with somewhat interesting theoretical results and a decent experimental study. I have some doubts about the novelty of the theoretical results. That is, prior works already seem to solve the problem of infectivity and continuity.

---

### Official Review · Reviewer_84dJ · 2022-10-22

**Confidence:** 3
**Correctness:** 3
**Technical Novelty And Significance:** 2
**Empirical Novelty And Significance:** 2
**Recommendation:** 3

**Clarity, Quality, Novelty And Reproducibility:**

The quality and clarity is mostly good.
However, the novelty and originality is limited.

**Strength And Weaknesses:**

## Strength

- This paper shows that there is no way to continuously embed a $M$-size set whose elements are from $R^d$ into less than $Md$ dimension with injectivity. I think formally claiming this is useful.

## Weakness

The significance of this paper is limited.

- This paper proposes an injective and continuous set representation in Eq (5) for $M$-size set with $R^d$ elements, and the final representation is $Md$ dimension. However, if we simply sort all elements in the set, and concatenate sorted elements together as a $Md$ vector, I believe this is also an injective and continuous set representation. Then, is there any significant benefit to use Eq (5)?
- From the experiment results, it seems that iceGNN-fixed is not feasible or worse than iceGNN-MLP in half of the cases, and improves slightly than iceGNN-MLP in the remaining half. It seems that injectivity is not very important in practice, but it is a key factor used to blame other models, e.g., GIN.
- Your iceGNN-fixed looks like a slight variant of EdgeConv which you can easily find in PyTorch Geometric library, thus I think the novelty of this is limited. Besides, it will be better to have GAT results.
- It will be a lot better if you can provide theory to study how dimension reduction hurts the injectivity and continuity since feature reduction is very common in real world applications.

Some statements are not precise.

- In the theorem 2, the definition of capital $D$ is missing. It is neither provided in appendix. Indeed, in the proof, the value of $D$ is different for different cases.




**Summary Of The Paper:**

This paper proposes an aggregation framework (set representation) which can lead to both injective and continuous GNN node embedding even if raw node features are from conintuous space rather than countable space.

**Summary Of The Review:**

This paper focues on both injectivity of continuity of GNN (set) representations on continuous space which is lack of formal studies.
Although this paper provides an interesting finding of requirement for such representation, the other parts including proposed methods and experinmental results does not give enough contribution.

---

### Official Review · Reviewer_eUaW · 2022-10-23

**Confidence:** 4
**Correctness:** 3
**Technical Novelty And Significance:** 3
**Empirical Novelty And Significance:** 3
**Recommendation:** 5

**Clarity, Quality, Novelty And Reproducibility:**

This paper is presented clearly, providing a practical method to construct injective aggregation functions along with its theoretical analysis.
As I mentioned in the above weaknesses, I have a bit of concern about its novelty. The presented learnable variant of the aggregation function is similar to the design in [1].


**Strength And Weaknesses:**

Strengths:
1. The theoretical analysis of the GNN and the injective functions are built on solid foundations,
2. This paper gives a practical method to construct injective aggregation functions.
3. The presentation logic is clear.

Weaknesses:
1. The experiments on node classification are not comprehensive enough. The three citation network datasets are rather toys, and the baselines are limited. More experiments on more datasets and baselines are expected. (Minor: Typo in Table4, ICGNN->iceGNN?)
2. My main concern is about the novelty. Considering the learnable variant of the aggregate function designed in this paper, it seems very similar to the original solution in [1], which resorts to an MLP (the universal approximation theorem) to model $f \circ \phi$ to get injective $\phi$.

[1] How Powerful are Graph Neural Networks?, Keyulu Xu, Weihua Hu, Jure Leskovec, Stefanie Jegelka, ICLR 2019

**Summary Of The Paper:**

This paper identifies the problem of non-injective functions when the dimensions of intermedia representation vectors are not enough. It proposes a new expressive GNN model with an emphasis on the injectivity and continuity of set functions.

**Summary Of The Review:**

 This paper provides a practical method to construct injective aggregation functions and solid theoretical analysis. However, I have concerns about the novelty of the proposed solution. I may raise my rating if my concern is resolved.

---

### Official Review · Reviewer_Bsrk · 2022-10-23

**Confidence:** 4
**Correctness:** 4
**Technical Novelty And Significance:** 3
**Empirical Novelty And Significance:** 2
**Recommendation:** 5

**Clarity, Quality, Novelty And Reproducibility:**

I really appreciate the authors' intellectual honesty in pointing the limitations of their results throughout the text ---which really helps with clarity. Overall, the main text is well written and easy to understand.  Code is available, which makes it as reproducible as we can ask from the authors.

Some minor comments:

a) Please, use proper punctuation when writing equations, e.g. periods or commas.
b) Please, add hyperparam search and choices for each architecture in the appendix.
c) Please, avoid inline text and sentences such as "it's easy to see from Lemma X [...]" in the proofs. It gets harder to read and some of the comments tend to not be mathematically precise. The more self-contained your work is, the better.

**Strength And Weaknesses:**

---Strength---
a) The problem tackled is relevant and few previous works have tried to address it.
b) As far as I was able to follow and check, all proofs and theoretical contributions hold.
c) The paper is well written and clarity is a strength.


---Weaknesses---
a) I'm not sure to what extent the changes in the functions proposed by the authors impact their own results, i.e. the functions in 4.2 vs the ones from the theorems. If it doesn't impact, which I'm pretty confident it's not true, then the authors should present the theory matching them. If it doesn't at least we would need to evaluate what's then the actual benefit of using their architecture.
b) The datasets used to evaluate the method are not sufficient to understand whether it outperforms existing solutions or not. We would need to see the method compared against more baselines, e.g. recent higher-order GNNs, in larger datasets, e.g. open graph benchmark (OGB).

**Summary Of The Paper:**

This work tackles the problem of using a injective and continuous set function as a neighborhood aggregation scheme in GNNs. The authors prove that such a construction exists (by extending some results from DeepSets) and how it would look like in GNNs. Finally, they present the practical benefits of such design in a set of benchmark tasks for graph learning.

**Summary Of The Review:**

At this point in the graph learning literature it's hard to accept a paper evaluated only on the TUDataset, Planetoid and ZINC benchmkarks. These were used five years ago and the field has progressed since then. I recommend that the authors take a look at OGB and compare against the baselines that are highly ranked there. Further, I recommend the authors elaborate a result for the gap between their theory and their architecture. My recommendation is thus a result of these weaknesses outweighing the paper's strength.

---

### Decision · Program_Chairs · 2023-01-20

**Decision:**

Reject

**Justification For Why Not Higher Score:**

The authors did not submit any author responses, making it hard to judge whether reviewer concerns are valid and solved.

**Justification For Why Not Lower Score:**

N/A

**Metareview: Summary, Strengths And Weaknesses:**

This paper proposes a theory for learning continuous and injective invariant graph functions. The previous expressivity study often ignores the continuity requirement and assumes the countability of input features. From this point, the motivation of the paper is good. However, there are several issues preventing the acceptance of the paper. First, as pointed out by the reviewers, the theorems have gap to the final models. For example, using MLP for dimension reduction loses the continuity gurantee. I would expect a full implementation of the theorems without dimensionality reduction and see how the continuity-preserving property influences the results. Second, the datasets used in the paper lack some recent ogb-level large ones. Thirdly, the authors did not submit author responses to address the reviewer concerns. For example, the simple layer proposed by Morris et al. (2019) is already injective and continuous.